# Kinetics and Kinematics of Working Trials Dogs: The Impact of Long Jump Length on Peak Vertical Landing Force and Joint Angulation

**DOI:** 10.3390/ani11102804

**Published:** 2021-09-26

**Authors:** Ellen Williams, Anne Carter, Jacqueline Boyd

**Affiliations:** 1Department of Animal Health, Behaviour & Welfare, Harper Adams University, Shropshire TF10 8NB, UK; ewilliams@harper-adams.ac.uk; 2School of Animal, Rural & Environmental Sciences, Nottingham Trent University, Nottingham NG25 0QF, UK; Jacqueline.boyd@ntu.ac.uk

**Keywords:** joint angulation, canine, biomechanics, peak vertical landing force, working trials, long jump

## Abstract

**Simple Summary:**

Working trials is a competitive canine discipline based on work originating from military and police dog work. Working trials competitions include dogs clearing a 9 ft long jump. Jumping over hurdle jumps or long jumps has the potential to cause injuries to the front limbs of dogs, and different jump heights can cause changes to landing forces and the angle of joints on landing. Little is known about the impact of the 9 ft long jump on landing force and joint angles in dogs. In this study, we aimed to determine whether altering the length of the long jump impacted dogs’ landing forces or joint angles. There was no relationship between the length of long jump and landing forces or joint angulation on landing, however, the greatest joint compression was observed on landing after traversing 9 ft. The dogs showed lots of individual variability. We recommend further research is undertaken to examine this individual variability and the effect of training and experience in working trials participants, to enable evidence-based recommendations for those training and competing dogs in working trials.

**Abstract:**

Working trials is a competitive canine discipline based on work undertaken by military and police dogs. A 9 ft long jump is a key component of the discipline. Research into landing forces and joint angulation in other canine disciplines has highlighted the potential for the occurrence of soft tissue injuries, predominantly in the front limbs. There is a paucity of work into the impact of spread/long jumps on joint angulation and peak vertical force (PVF) on landing, and limited research on working trials dogs generally. This study aimed to determine whether altering the length of the long jump impacted PVF and apparent joint angulation upon landing. 21 dogs regularly competing in working trials cleared the long jump at three lengths: 9 ft (full length), 8 ft, and 7 ft. The impact of altered long jump length on the PVF, apparent shoulder and carpus angulation, and duration of landing, were analysed using general linear mixed models. There was no significant relationship between the length of the long jump and PVF or joint angulation on landing (*p* > 0.05). Greatest joint compression was observed on landing after clearing 9 ft. Individual variability in landing joint angulation, PVF and force distribution of the left and right front limbs on landing was observed across all three experimental lengths. We recommend further research is undertaken to examine individual variability and the effect of training and experience in working trials participants, to provide evidence-based recommendations for training people and competing dogs in this discipline.

## 1. Introduction

Working trials originated as a competitive canine discipline in the 1920s, based on tasks required by military and police dog work. Working trials competitions require dogs to complete three components: scent work, agility (clearing a 6 ft scale (wall), 9 ft long jump and a 3 ft hurdle, all ‘under control’), and ‘control’ obedience tasks [1]. The long jump consists of five sections, no less than 3 ft (91 cm) wide, of increasing width and height, from 4 inches (101.6 mm) high rising to 7 inches (177.8 mm) at the rear edge. The total distance of the long jump is 9 ft (274 cm). By comparison, the long jump in Kennel Club (KC) agility competition consists of three to five sections, with an overall length of between 1.2 m and 1.5 m for large dogs (>43 cm at the wither) [2], a height of 127 mm at the front to 381 mm at the rear. The working trials long jump also differs from the agility long jump by the absence of upright, 1.2 m high marker poles at each corner of the obstacle, which are seen in agility [2]. Whilst rarely used in agility competition, the only other obstacle requiring the dog to jump over distance (i.e., a spread jump) rather than height, is the water jump, ranging from 40–50 cm for small dogs and up to 1.2 to 1.5 m for large dogs [2]. 

The kinematics of dogs traversing upright hurdles have been investigated in terms of jump height [3,4] and distance between obstacles [5,6]. However, the long jump requires the dogs to jump a much greater distance and therefore is reliant on forward velocity rather than height. It thus requires an altered jumping technique in dogs, compared to that employed to clear an upright hurdle [7]. In horses, fence height and width did not affect limb placement in horses on take-off but horses appeared to adjust their landing [8]. When clearing a water jump of 4.5 m, vertical velocity was positively correlated with distance jumped in horses [9]. Landing forces experienced by a locomoting or jumping animal can be indicated by measuring the peak vertical force (PVF), which represents the single maximum force exerted perpendicular to the ground surface during a stance phase of locomotion. Comparisons between the PVF experienced by dogs traversing a hurdle jump versus a long jump, found a higher PVF when landing from a hurdle jump [7]. More loading occurred on front limbs than rear when traversing the long jump (57%) although marginally less than the hurdle (60%) [7]. However, this comparison was made between dogs jumping heights and distances encountered in agility rather than the increased distance (×1.8) found in the working trials long jump obstacle. Furthermore, this was based on jumps whereby the height and length of the jump are adjusted according to the category in which the dog’s height to withers falls, as opposed to the long jump in working trials which is a set distance for all competitors. The impact of the length of the long jump on the kinematics and kinetics of working trials dogs is unknown. 

Injury risk in dog sports has been primarily examined in agility and flyball dogs [10,11,12,13,14,15]. The biomechanical factors associated with injury risk in sporting dogs were indicated as a key research area by Cullen et al. [11] following their retrospective survey of injury risk in agility dogs, where increased odds of injury were associated with less experienced dogs. Kerr et al. [12] highlighted that front paw injuries were most common (23.7%) followed by shoulder injuries (15.8%) from a survey of agility dogs and handlers, but also suggested that an increased dog age was linked to increased injury risk. Flyball is a canine discipline where a sequence of upright hurdles is traversed at speed. Montalbano et al. [14] reported that younger flyball dogs had an increased injury risk where limb injuries were the most common, especially the forelimbs. Conversely, increased age of dog was associated with increased injury risk in flyball dogs [15], although forelimbs again represented the areas most injured. The increased distance of the long jump in working trials compared to agility has the potential to increase both peak vertical landing force and compression of joints on landing and therefore injury risk at the maximum distances. Whilst agility competition allows hurdle jump heights to be lowered, and long jump distances to be shortened for smaller dogs (based on height to withers), working trials requires all dogs, regardless of size, to complete the same height and length of obstacles.

While researchers have investigated the jump competence within agility-based competitions, there remains a lack of research on the impact of spread jumps (long jump and water jump in agility and long jump in working trials) on PVF and apparent joint angulation of dogs on landing. Research into the impact of vertical jumps has indicated impacts on the physical health of dogs and highlighted the different jumping techniques adopted by lesser and more experienced dogs [16]. Furthermore, work by Carter et al. [17] identified changes in peak vertical landing force and joint angulation in dogs traversing different heights in the working trials scale obstacle. Beyond the work of Pfau et al. [7], to the authors’ knowledge, there is no research on the impact of spread jumps on agility dogs, and there is no publicly available research on the impact of the length of long jump on dogs regularly competing in working trials. 

This study aimed to determine whether an alteration in long jump length impacted the peak vertical force (PVF) and apparent joint angles on landing in dogs routinely training and competing in working trials. This will enable a greater understanding of the impact of this obstacle on dog health and permit evidence-based advice to support training and competitive standards. 

## 2. Materials and Methods

### 2.1. Study Population

Dogs were recruited opportunistically from those regularly training and competing in working trials in the UK. Contact was made with UK working trials representatives via email and interested members of corresponding groups were invited to participate in the study. To be included in the study dogs had to meet the following criteria: aged over 12 months, had been regularly training for and/or competing in working trials competitions for at least 12 months, and were physically fit enough to take part in the research project. All dogs had competed in working trials for at least one year and were therefore experienced in clearing the long jump obstacle at the maximum distance. This minimised the effect of naive or inexperienced dogs. 

Demographic details of the study population are provided in Table 1. 21 dogs (15 male, 6 female) were recruited to the study from five breeds/types, as described by their handlers; border collie/working sheepdog (10), golden retriever (1), German shepherd/malinois (4), Labrador retriever (5), spaniel cross Labrador (1). 

### 2.2. Experimental Setup

The study was carried out in a fenced outdoor equestrian arena with a fibre sand surface. Handlers prepared their dogs as they would in working trials competitions, allowing acclimation to the research environment and equipment. The study examined dogs traversing the long jump at three different lengths—9 ft (2.74 m) (the current maximum KC length in competition), 8 ft (2.44 m), and 7 ft (2.13 m). Dogs were directed by their usual handler throughout the study. Dogs traversed the long jump as per a normal training or competition session. They were asked to complete the long jump exercise three times per distance. Where dogs did not land fully on the pressure sensing equipment, they repeated the jump to achieve three successful landings on the mat. The number of times each dog traversed the long jump is included in Table 1. The order of the three distances was randomised. No time limit was put on completion of the obstacle; therefore, breaks could be taken between attempts. If a dog failed to complete a long jump, they were given one further attempt at that distance, following a second failed attempt, the dog was withdrawn from the study. Dogs were withdrawn from the study at the owner’s discretion (e.g., dog appeared reluctant to continue with the exercise). Dogs were filmed during the landing phase of clearing the long jump using high-definition video cameras (JVC-GC PX 10 HD, 300 fps, JVC, Watford, UK) with lateral placement to the long jump with a one-metre ground marker for reference (Figure 1).

#### 2.2.1. Peak Vertical Landing Force 

Peak vertical landing force (PVF) was measured using a Tekscan walkway gait analysis system 3150 pressure sensing mat (sensing area 0.87 × 0.37 m^2^) (Tekscan, Biosense Medical, Little Waltham, UK). This was positioned at the landing point (Figure 1) and covered by a thin rubber mat to standardise the landing surface. Peak vertical force on landing across both front feet was measured using Matscan XL (Tekscan, Biosense Medical, Little Waltham, UK) (Figure 2). Where only one foot landed on the mat replicates were discarded. 

#### 2.2.2. Apparent Carpus and Shoulder Angles and Duration of Landing

Apparent carpus and shoulder angles, and duration of landing were measured using Kinovea Version 0.9.3. Apparent carpus and shoulder angles were identified based on the visualized anatomical landmarks of shoulder and carpus from video footage (Figure 3) and were measured on each video frame (30 frames per second) during the landing phase, from the time the first front foot touched the floor to the time the first rear foot hit the floor. The right forelimb was measured in the first instance with the left forelimb utilized if the angles of the right forelimb could not be clearly measured. Forelimb choice was consistent within the dog. The frame at which the dog had the minimum carpus angle was taken to be the lowest phase of the landing. Minimum carpus angle, shoulder angle at the lowest phase of the landing and minimum shoulder angles were used for analysis. Duration of landing was measured in seconds (using video frames). 

### 2.3. Data Analysis

Data analysis was undertaken as per Carter et al. [17]. Linear mixed models, with Tukey corrected post-hoc tests where appropriate, were used to investigate the impact of the length of the long jump (7 ft, 8 ft, 9 ft) and dog bodyweight (<25 kg, >25 kg), on peak vertical landing force, duration of landing (seconds), minimum carpus and shoulder angles on landing. Six models were created: PVF, PVF as a function of bodyweight, duration of landing (number of data frames, recorded at 30 FPS) carpus angle at lowest phase of the landing (minimum carpus angle), shoulder angle at lowest phase of the landing and minimum shoulder angle. To prevent erroneous identification of peak vertical landing force, individual jumps were only included in the analysis if values were present for both front feet, to enable identification of the highest landing force across both feet. Peak vertical landing force and angles of interest were fitted as response variables. Length of the long jump and dog weight were fitted as fixed effects. To control for replicates, the dog was included as a random effect in each model. Data analysis was undertaken in R Studio (Version 4.0.3) [18] using packages ‘lme4’ [19] and ‘emmeans’ [20]. Variance in PVF, PVF as a function of body weight, apparent angles on landing and landing duration between <25 kg and >25 kg at the three lengths (7 ft, 8 ft, 9 ft) was assessed using a Levene’s test using package ‘car’ [21]. Graphs were produced using package ‘ggplot2’. Model results are reported as a model estimate (β1) ± SE. Significance values were set at *p* < 0.05 for all analyses.

## 3. Results

### 3.1. Demographic Details of Study Dogs

15 male dogs and 6 female dogs made up the study population. The median age of dogs was 4.5 years (range 2–8 years). Dogs <25 kg (*n* = 12) had a mean ± SD bodyweight of 21.5 ± 2.4 kg, dogs >25 kg (*n* = 9) had a mean ± SD bodyweight of 29.2 ± 4.3 kg.

### 3.2. Peak Vertical Landing Force (PVF)

There was no difference in peak vertical landing force for dogs of either <25 kg or >25 kg at any of the three lengths (*p* > 0.05) (Figure 4). There was a trend towards greater variation (Figure 5) in PVF in dogs < 25 kg (*p* = 0.07) and there was a trend towards dogs <25 kg having higher PVF than dogs >25 kg (−6.458 ± 3.408, Z = −1.895, *p* = 0.0581). 

Peak vertical landing force as a proportion of dog body weight was variable across dogs. Lighter dogs (<25 kg) had a greater PVF in proportion to their body weight than dogs >25 kg (−0.44 ± 0.18, t = −2.422, *p* < 0.05). There was greater variation in PVF as a proportion of body weight in dogs <25 kg (F = 2.8542, *p* < 0.05) (Figure 5). 

#### Individual Variation

Individual variation was observed in terms of equality of peak vertical force on each paw on landing. The difference between mean PVF on the two front feet in the study dogs is displayed in Table 2. Descriptively, it appears that some dogs have greater variation in the landing pressure on each foot (e.g., dog 2 and dog 21) and others show more consistency (e.g., dog 5), indicating that some dogs may exert more force on one particular limb than those who land more symmetrically. There was no significant difference between the difference in peak vertical landing force as a function of the length of long jump (*p* > 0.05) or bodyweight of the dog (*p* > 0.05). There was no significant variation between the difference in PVF between front feet for dogs <25 kg or >25 kg (*p* > 0.05) (Figure 6).

### 3.3. Landing Duration

Duration of landing was calculated as the number of data frames (recorded at 30 FPS). There was no significant difference in duration of landing for dogs of either <25 kg or >25 kg at any of the three lengths (mean frames ± SD; 7 ft: 9.9 ± 3.4; 8 ft: 9.6 ± 2.3; 9 ft: 9.4 ± 3.1). There was no significant difference in variation of landing duration between dogs <25 kg or >25 kg at any of the lengths. 

### 3.4. Apparent Carpal and Shoulder Angles on Landing

Dogs that were <25 kg had smaller minimum carpal angles on landing than dogs >25 kg (7.484 ± 3.211, Z = 2.331, *p* < 0.05), however, there was no relationship between this and the length of the jump (*p* > 0.05). There was no difference in variation of minimum carpal angle in dogs <25 kg or >25 kg (Figure 7). 

Dogs that were >25 kg had smaller minimum shoulder angles on landing than dogs <25 kg (−9.831 ± 3.482, Z = −2.559, *p* = 0.0105). However, there was no relationship between apparent shoulder angle and length of the jump (*p* > 0.05). There was no difference in variation of minimum shoulder angle in dogs <25 kg or >25 kg (Figure 8). 

There was no significant difference between shoulder angle at the lowest phase of the jump (i.e., at minimum carpal angle) at any of the lengths for either dogs <25 kg or >25 kg. However, there was significantly more variance in shoulder angle at the lowest phase of the landing; dogs >25 kg showed greater variation than dogs <25 kg (F = 2.8904, *p* = 0.0154) (Figure 9). 

## 4. Discussion

Working trials is a canine discipline that requires dogs to undertake several agility components, including clearing a 9 ft long jump. Whilst the impact of hurdle height in agility dogs is relatively well researched, there is a paucity of literature on the impact of the agility components of working trials on participating dogs. Research conducted by the project team [17] highlighted that there may be benefits to dogs in reducing the height of the scale obstacle for dogs in working trials. The impact of spread jumps in canine agility (such as the long jump and water jump) and the long jump in working trials has been little studied. However, research examining equine jumping styles suggests there are likely to be variations in impact on peak vertical landing force and joint angulation on landing according to the length/spread of the jump [8,9]. The long jump in canine working trials is 1.8× longer than the greatest distance in canine agility competition [1,2] and thus, there is the potential to impact upon working trials dogs. 

This piece of research aimed to determine whether an alteration in long jump length altered the peak vertical landing force and apparent carpal and shoulder angles of dogs on landing. The impact of bodyweight of dogs (split into two weight categories of <25 kg and >25 kg) was also assessed, to determine the effect of approximate dog size on landing metrics. Dogs showed considerable variability in both their peak vertical landing forces and apparent joint angulation after clearing the obstacle. 

### 4.1. Peak Vertical Landing Force

Pfau et al. [7] highlighted greater loading on the front limbs than rear limbs of agility dogs after clearing the long jump and research into injury risk of dogs competing in flyball has highlighted the forelimb as the predominant point of injury [14,15]. The front limbs were the areas of focus for this piece of research. In this study, there was a trend towards heavier dogs (bodyweight) showing greater peak vertical landing force than lighter dogs. When this was investigated in terms of the landing force in relation to individual body weight, lighter dogs displayed greater landing force than heavier dogs. This was the reverse of the findings from the same study population in relation to peak vertical landing force when traversing the scale obstacle [17]. However, the scale is a relatively static obstacle, with dogs landing and then remaining in a stationary position, before returning, back over the obstacle. In contrast, the long jump requires the dog to clear the length of the jump and continue to move forwards with high forward velocity. This provides the opportunity for the dog to take a more fluid approach to clearing the obstacle, and it could thus be considered beneficial for dogs to take a flatter and lower jumping trajectory than for an upright obstacle. In agility, it is presumed that the flatter the jump, the more energy is spent in forwarding movement and the less energy is wasted in upward movement [22]. The differences seen as a proportion of dogs’ body weight may indicate two different jumping techniques employed by dogs, with heavier dogs ‘landing’ before continuing, and some lighter dogs running immediately off the obstacle, effectively as a continued stride. There was however no significant difference in duration of landing between dogs of different weights which suggests the point of landing was still relatively transitory, even in the heavier dogs. Notably, analysis of digit injuries in agility dogs identified that a greater weight-to-height ratio was associated with an increased injury risk [13], further suggesting that bodyweight and morphometrics could account for the variation observed.

#### Individual Variation

Due to the fluid nature of how the long jump is traversed, it was anticipated that dogs would show variation in peak vertical landing force across the two front feet, rather than the more equal distribution of PVF which may be observed in dogs landing from a static height (e.g., a car boot or working dog trials scale). 

In horses, it is the trailing forelimb that takes the majority of the force, with estimates of up to 2.6× body weight on that limb [23]. Although overall, there was no statistically significant variation between the difference in peak vertical landing force for the front feet for the study dogs, descriptively it appeared that some dogs may be exerting more force on one particular limb as opposed to those who were landing more symmetrically. Dogs often show a preference for ‘handedness’ [24]. Research into lateralisation in dogs has suggested that dogs with a weaker paw preference show greater distractibility in agility [25] and left paw departure laterality has been linked with improved performance when dogs are performing tasks that involve forward movement [26]. 

As dogs only performed approximately three replicates at each long-jump length it was beyond the scope of this study to investigate the impact of paw preference in terms of take-off and landing. However, investigating whether dogs consistently exhibit a foot preference on landing and how this relates to departure laterality would enable a greater understanding of the likelihood of excessive strain on a particular joint or limb and thus potential injury risk. It would also be interesting to further understand the relationship between paw departure laterality in obstacle clearance and paw departure laterality when loose lead walking, to understand whether dogs change their approaches in different settings. 

Investigation of individual differences in canine athletes is recommended, principally we suggest the following: determination of whether there is a difference between landing and trailing forelimb, determining whether dogs consistently exert more force on one particular limb than another on landing, identification of factors which affect that landing force distribution and investigation into whether there are specific aspects of each front foot (e.g., front pads, side pads) that experiences highest pressure loads.

### 4.2. Apparent Joint Angulation on Landing

Previous research has indicated that dogs will alter joint angles to reduce the impact of force when landing [7]. Our findings support this concept. The predominant joint in which force was taken varied across dogs; heavier dogs showed more compression in the shoulder angle whereas lighter dogs showed more compression in the carpal joint. These findings mirror results that were found in the scale obstacle [17], which indicates that dogs are controlling their own landings. In both instances, the level of this compression was not significantly affected by the length of the long jump. However, whilst not significant, angles were smallest on the longest jump (9 ft), indicating greatest compression, which again replicates what was seen in the same study dogs traversing the scale obstacle [17]. There was substantial individual variation in the study dogs and so it is likely that individual dogs are adapting their jumping style accordingly. There may also be an impact of training and handling style on how dogs traverse equipment in a range of contexts. This was not examined in the current study, although represents an interesting area for future examination. 

Consideration should be given to reducing the frequency with which dogs train over the full distance of the long jump, in recognition of the potential implications on landing joint angulation for some dogs and the potential for this to contribute to injury risk in the front limbs. Anecdotally, owners reported that it was on the long jump that dogs ‘showed signs of age’, potentially indicative of ‘wear and tear’. It is therefore also recommended that further work is undertaken on dogs of a wider age range, to explore the effect of age on performance and kinematics, and to limit the potential detrimental impact of older dogs traversing the 9 ft jump.

### 4.3. Obstacle Clearance

Successful obstacle clearance relies on dogs assessing and executing an optimal combination of both forward velocity and distance to obstacle [7,22]. Dog experience can impact the suspension phase of the jump seen in agility dogs, with alterations seen to the optimisation of take-off and landing phases [22]. It has been proposed that more experienced dogs should execute long and quick jumps [22]. The experimental set-up did not enable accurate data to be garnered on the suspension phase of the jump, nor the distance of landing from the obstacle. However, anecdotally we observed dogs altering the way they jumped according to the length of the jump, with dogs only jumping the distance required for the obstacle, rather than overjumping to the standard 9 ft distance. This was despite participating handlers widely indicating that they expected their dogs to jump the full length, whether the obstacle was that or not. This phenomenon of adaptation has also been seen in horses [27]. All the dogs who took part in this study were experienced, having been regularly training and/or competing in working trials for at least 12 months. Alterations in jumping style suggest that dogs were accurately perceiving the length of the long jump and reducing unnecessary energy expenditure/injury risk. However, further work should be undertaken on this obstacle to assess this theory. Consideration of the level the dogs have reached in competition may provide an extra opportunity to measure the effect of training and previous experience on dog jumping styles, as is evident in agility dogs [16,22]. and to understand the potential impacts on dogs who are new to working trials. 

### 4.4. Future Directions for Working Trials

The welfare of sporting dogs has been advocated as an important area for future research [11,28] and we also recommend building on this preliminary research to further enhance our understanding of the impact of training for and competing in working trials. 

It was beyond the objectives of this study to investigate the impact of solidness of landing surface on PVF and joint angles on landing, and there is the potential for this to have an impact. Whilst every effort was made to provide a true replicate of the types of environments working trials may be undertaken in, and the experiment was not undertaken in a laboratory setting, dogs in working trials compete on many different types of terrain. It is therefore recommended that further research is undertaken in-situ, during working trials, to capture the range of conditions dogs may be working and training in before a recommendation can be made on appropriateness of the length of the long jump. 

Finally, active working dogs should not carry excess bodyweight and those engaging in competitions should minimise excess bodyweight to reduce loading on joints. Consideration should also be given to the impact of dog weight and build, as well as height-to-withers. Traditional measurements for categorising agility dogs in the UK focus on height to withers. Whilst this is an invariable measurement, it does not account for ‘heavier-set’ dogs who may not be particularly tall and may be exerting extra forces on joints during movement and jumping. Considering dog bodyweight could enable modification of jumps and obstacles where required. In relation to working trials, we recommend that thought is given to the impacts of altering the length of the long jump based on canine morphometrics, in the same way as agility obstacles are altered. Thought could also be given to developing differential competitive categories for novice, veteran or dogs otherwise less suited to traversing obstacles according to current working trials regulations in the UK. 

## 5. Conclusions

The welfare of sporting dogs has been advocated as an important area for future research. Taking an evidence-based approach to the development of guidelines in relation to long jump length in training and working trials competitions is important to ensure minimum negative impacts on the health and welfare of participating dogs. Whilst there was no significant relationship between the length of the long jump and PVF or joint angulation on landing, the greatest compression in the joints was observed at the longest distance (9 ft). Dogs in this study showed considerable variability in joint angulation and peak vertical forces on landing over varying distances of the long jump. Furthermore, individual variability was seen in force distribution on the left and right front limbs on landing. Previous research of injury in agility and flyball dogs has highlighted front limbs as being the main point of injury. Injury risk is also affected by the experience. The dogs assessed as part of this research were all experienced in working trials, which may have led them to employ jumping techniques that minimise injury risk. The high variability seen in the dogs in this study is something that warrants further investigation, and we advocate further research into individuality in canine athletes in relation to this aspect of working trials. Indeed, a limitation of this study is the extent of individual variability observed within participating dogs, including biological parameters such as age, sex, bodyweight, breed/type, and body size. There will also be a likely impact of variation between training techniques, experience, and level of fitness within this study population. For this reason, we recommend building on this preliminary research to consider the impact of age/experience in a wider range of dogs from the working trials population. This will enable and support evidence-based recommendations for training people and competing dogsin this discipline. We also recommend that further research is undertaken to determine whether offering young/inexperienced and older dogs alternative distances as part of accommodating their individual needs and whether distances should be relevant to dogs’ height to withers and/or weight, as is common practice in agility competitions, would help to reduce injury risk for dogs training and competing in this discipline.

## Figures and Tables

**Figure 1 animals-11-02804-f001:**
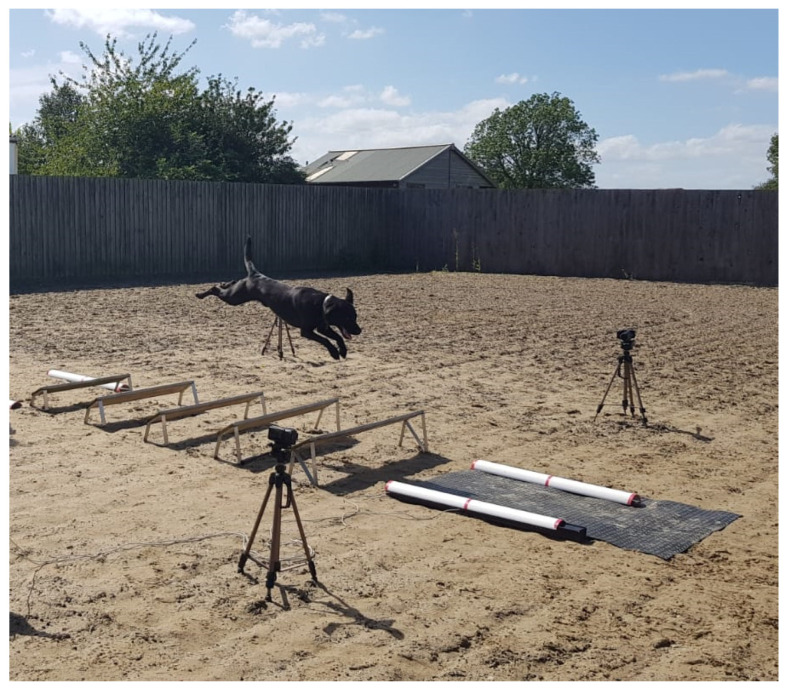
Long jump study setup showing the positioning of pressure sensing mat (under protective rubber sheeting) on the landing side.

**Figure 2 animals-11-02804-f002:**
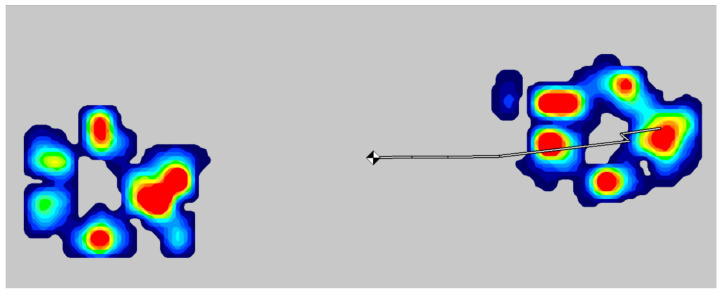
Tekscan ‘heatmat’ visualisation of peak vertical landing force (N). Colours provide a visual representation of measured forces from low (blue) to high (red). The black and white symbol indicates the centre of gravity at the point of measurement.

**Figure 3 animals-11-02804-f003:**
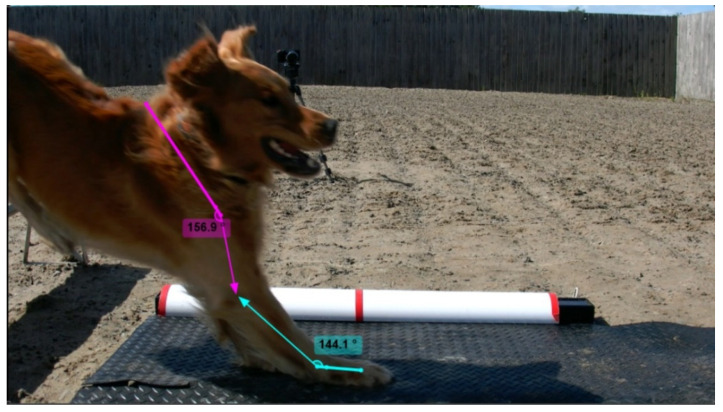
Apparent angles of the carpus (green angle line) and shoulder (pink angle line) of dogs on landing after clearing the long jump, measured using Kinovea Version 0.9.3 (www.kinovea.org.uk, accessed on 10 June 2021).

**Figure 4 animals-11-02804-f004:**
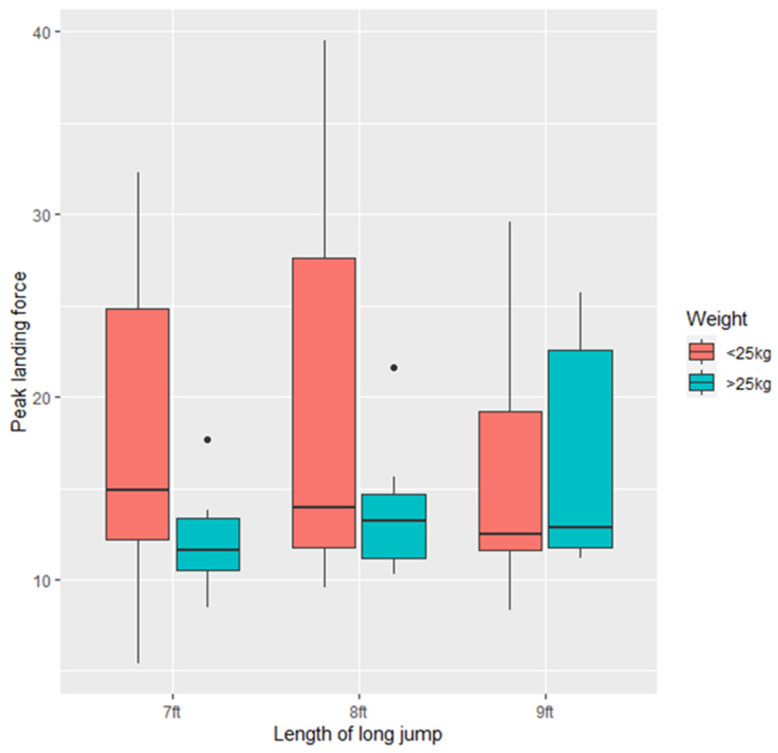
Maximum peak vertical landing force (lbs) for the study dogs (*n* = 21) at the three long jump lengths (7 ft, 8 ft, 9 ft).

**Figure 5 animals-11-02804-f005:**
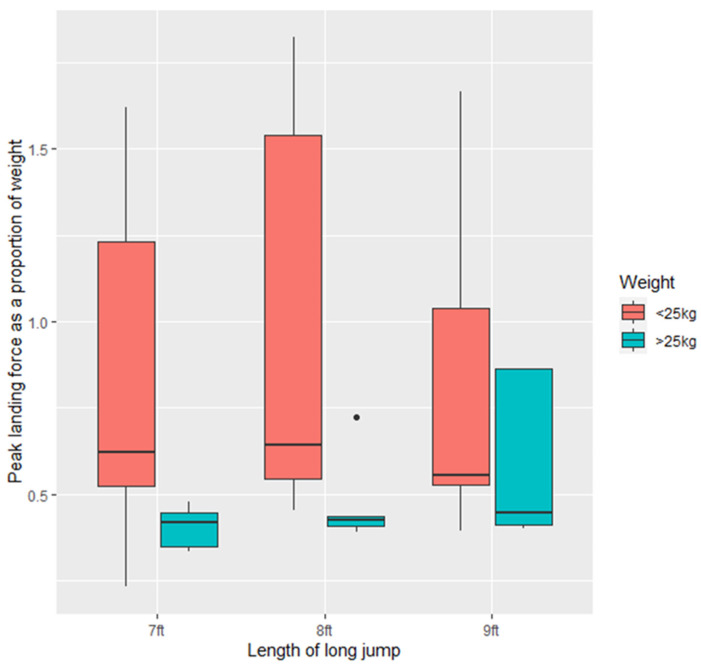
Peak vertical landing force (PVF) as a proportion of body weight (lbs/kg) across all the study dogs (*n* = 21) at the three long jump lengths (7 ft, 8 ft, 9 ft).

**Figure 6 animals-11-02804-f006:**
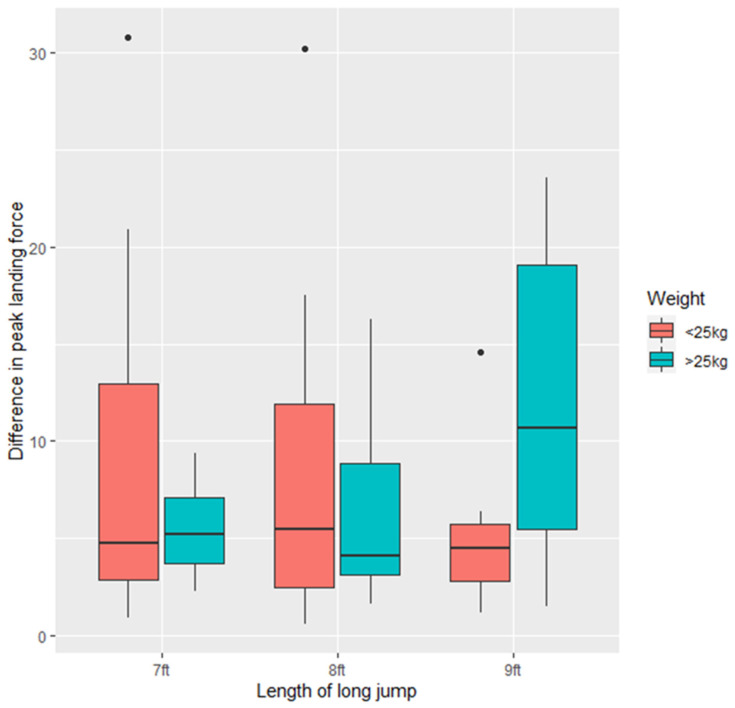
Difference in peak vertical landing force (lbs) between the two front feet for the study dogs (*n* = 21) at the three long jump lengths (7 ft, 8 ft, 9 ft).

**Figure 7 animals-11-02804-f007:**
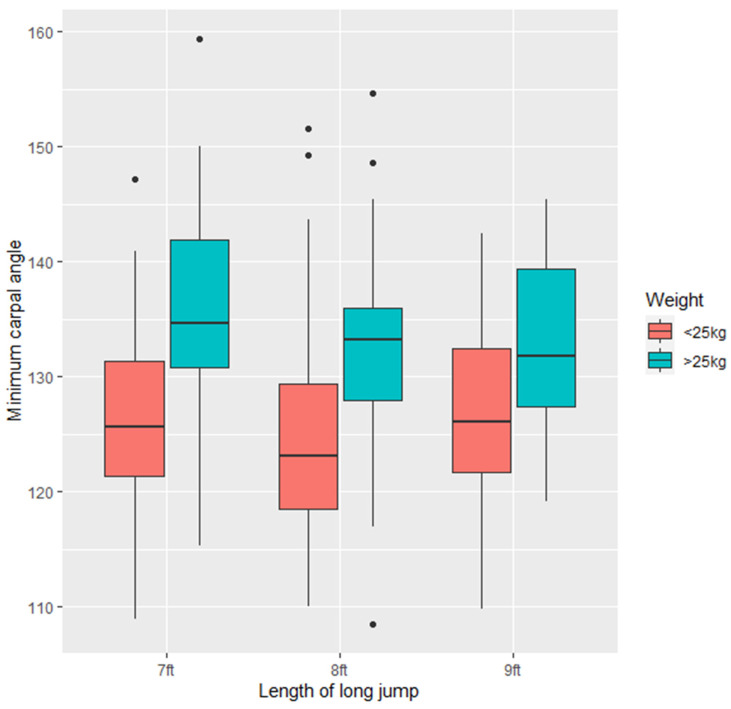
Minimum apparent carpus angle (degrees) across all the study dogs (*n* = 21) at the three long jump lengths (7 ft, 8 ft, 9 ft).

**Figure 8 animals-11-02804-f008:**
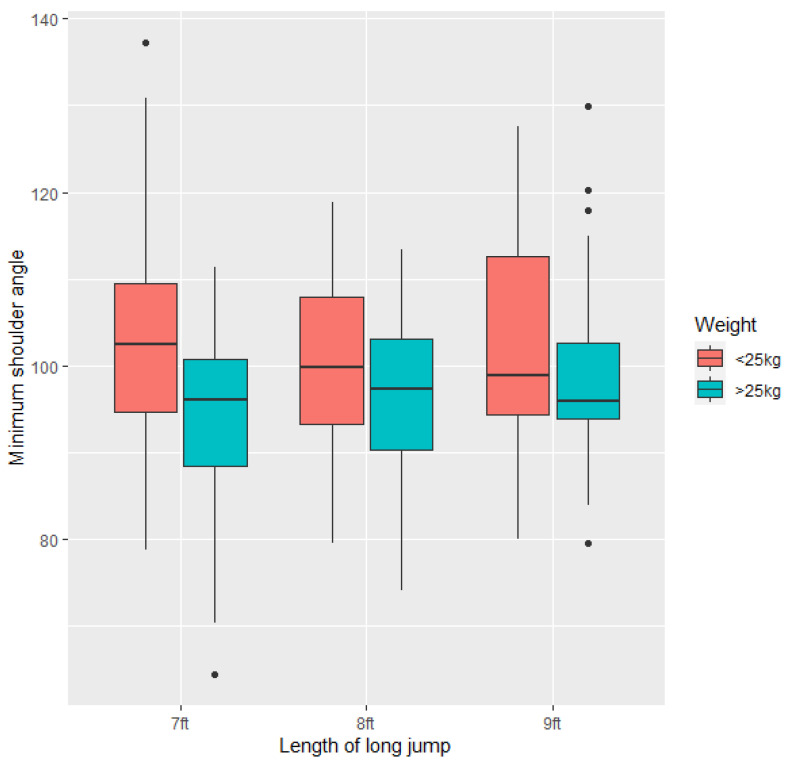
Minimum apparent shoulder angle (degrees) across all the study dogs (*n* = 21) at the three long jump lengths (7 ft, 8 ft, 9 ft).

**Figure 9 animals-11-02804-f009:**
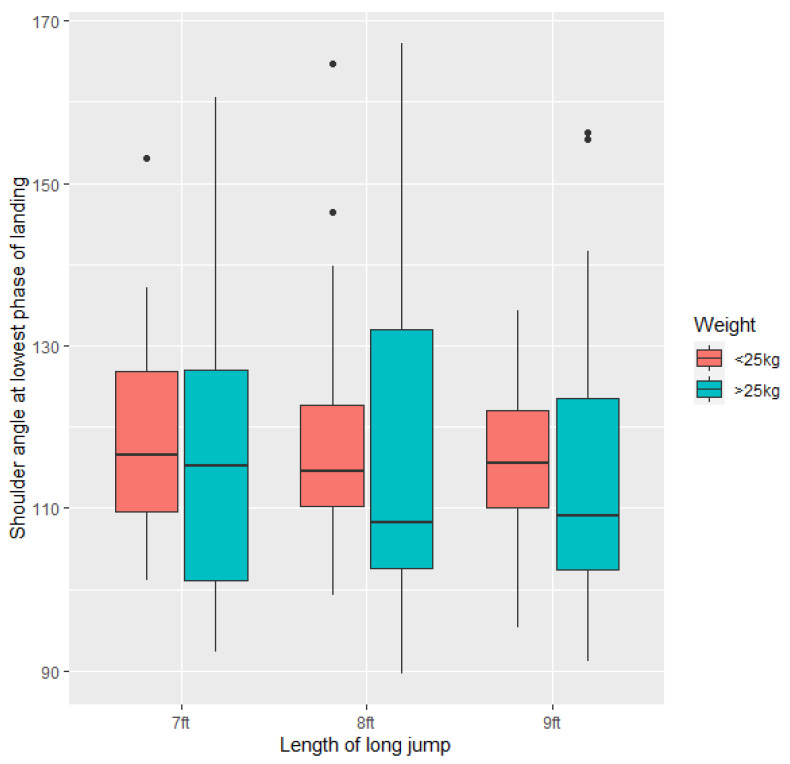
Apparent shoulder angle (degrees) at the lowest phase of landing across all the study dogs (*n* = 21) at the three long jump lengths (7 ft, 8 ft, 9 ft).

**Table 1 animals-11-02804-t001:** Demographics of participating dogs (as per those detailed in Carter et al. [17]).

Dog	Sex	Breed/Type	Age (yr)	Height to Withers (cm)	Weight (kg)	Number of Long Jump Completions *		
						7 ft	8 ft	9 ft
1	Male	Working sheepdog	8	59.0	23.7	3	3	3
2	Male	Border collie	4	49.5	21.7	3	3	3
3	Male	Labrador retriever	5	57.0	27.7	5	3	4
4	Female	Labrador retriever	3	51.0	21.0	3	3	5
5	Male	Working sheepdog	3	56.5	24.1	4	4	3
6	Male	Border collie	7	56.5	23.3	4	0	1
7	Male	Working sheepdog	8	53.0	17.8	3	3	3
8	Male	German shepherd	6	65.0	40.0	3	3	6
9	Female	Labrador retriever	5	55.0	24.6	4	4	5
10	Female	Border collie	2	48.3	17.2	3	5	3
11	Male	Labrador retriever	5	57.0	31.3	5	4	6
12	Male	Labrador retriever	4	57.0	25.8	4	4	4
13	Male	Spaniel/Labrador retriever cross	6	47.0	23.2	3	0	2
14	Male	Border collie	3	56.0	25.2	9	0	0
15	Female	German shepherd	5	No data **	30.2	4	2	1
16	Female	Working sheepdog	3	52.0	18.6	4	4	3
17	Male	Border collie	2	55.0	21.3	3	6	5
18	Female	German shepherd	3	56.5	26.3	5	4	3
19	Male	Malinois	No data **	54.0	26.6	4	3	5
20	Male	Working sheepdog	7	52.0	21.7	3	3	3
21	Male	Working golden retriever	4	57.0	29.8	5	3	4

* Dogs traversed the long jump distance height until they were considered to have landed successfully on the pressure mat three times (visual assessment from the project team). Continued traversing of the long jump to achieve three successful landings on the pressure mat was at the discretion of the handler. ** Where no data was collected, this was due to difficulty in accurately measuring height.

**Table 2 animals-11-02804-t002:** Mean difference (±SD) between peak vertical landing force on dogs two front feet. PVF was only calculated when both feet successfully landed on the mat (and recorded maximum landing force on Tekscan mat).

Dog	Number Successful Replicates	7 ft	Number Successful Replicates	8 ft	Number Successful Replicates	9 ft
1	2	10.0 ± 5.3	1	10.5 ± 0.0	1	6.4 ± 0.0
2	1	30.8 ± 0.0	2	22.2 ± 8.0	0	-
3	2	5.4 ± 1.7	1	4.1 ± 0.0	1	1.5 ± 0.0
4	3	3.7 ± 0.6	3	4.6 ± 1.6	3	4.4 ± 1.6
5	2	2.5 ± 0.6	3	3.0 ± 1.2	2	4.4 ± 1.2
6	2	6.8 ± 4.6	0	-	1	6.2 ± 0.0
7	3	10.1 ± 6.6	3	6.9 ± 7.5	2	1.4 ± 0.1
8	1	9.4 ± 0.0	1	11.4 ± 0.0	0	-
9	2	5.1 ± 1.6	1	5.5 ± 0.0	0	-
10	2	10.5 ± 4.2	3	9.0 ± 6.1	2	9.7 ± 4.9
11	2	4.4 ± 2.1	2	2.7 ± 1.1	0	-
12	1	4.2 ± 0.0	1	6.3 ± 0.0	0	-
13	3	3.0 ± 1.3	0	-	1	2.7 ± 0.0
14	2	4.4 ± 0.9	0	-	0	-
15	0	-	0	-	1	10.7 ± 0.0
16	0	-	0	-	0	-
17	1	20.9 ± 0.0	0	-	0	-
18	0	-	1	2.6 ± 0.0	0	-
19	1	8.3 ± 0.0	0	-	1	5.5 ± 0.0
20	1	6.2 ± 0.0	1	6.3 ± 0.0	2	4.0 ± 0.3
21	0	-	1	16.3 ± 0.0	2	21.4 ± 2.2

## Data Availability

Data available from the corresponding author upon reasonable request.

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
