# Peer review of "Kinetics and Kinematics of Working Trials Dogs: The Impact of Long Jump Length on Peak Vertical Landing Force and Joint Angulation"

_animals, 2021, doi:10.3390/ani11102804_

Round 1

Reviewer 1 Report

Interesting research, given the ever increasing number of dogs that partecipate in sporting competitions and which require an athletic performance not always adequate to their characteristics and training. Studies aimed at protecting the welfare and health of these dogs are needed. The authors well explain that this is a preliminary study and that further investigations will be needed to rule out the numerous variables that can be taken into consideration.  I think the authors should describe in more detail what they mean by trained dogs and what the training programs were. The authors should better define the type of lesion found according to the size of the dogs, the breed and the training methods.

Author Response

please see response to comments in the attached document

Reviewer 2 Report

Dear authors,

I find your work really interesting. As you mention, you deal with a current and not too much studied topic, so your work could be useful in the improvement of our working dogs life’s quality and welfare.

I consider that the article is properly structured and written, and attached tables and images help in the understanding of it.

Additionally, you should take into account these considerations that, in my opinion, will improve the quality of the article:

  • In the introduction explain briefly what the peak vertical force (PVF) is.
  • Line 5: I assume that "KC agility" is "Kennel Club agility", but you must specify the words before including the abbreviation.
  • Explain in detail how dogs included in the study were selected.
  • Why were 21 dogs included in the study? Is it a statistically representative number of the population?
  • Specify inclusion and exclusion criteria for the selected dogs.
  • In this study, males predominate more than females. Is there a difference between the sexes for this practice?
  • Is there a difference between the older dogs that exerted more pressure on the platform? I have seen that you have differentiated between weights, but have you studied if the age of the animal influences in any way?
  • Line 146: “Dogs were also withdrawn from the study at the owner’s discretion”. What were the reasons the owners removed the dogs from the study?
  • Line 171: specify which are the anatomical landmarks
  • In the figures please clarify what symbols mean (.)
  • Line 366 you indicate that: “heavier dogs showed more compression in the shoulder angle whereas lighter dogs showed more compression in the ankle”. Do you mean the ankle or the wrist?
  • In the conclusions of the article, I recommend adding a paragraph that specifies the limitations of the study (such as the one you name of the variability between individuals)

Author Response

(The authors gave the same response as above.)
